# Using Synchronous Fluorescence to Investigate Chemical Interactions Influencing Foam Characteristics in Sparkling Wines

**Bruna Condé [1]** , **Alanna Robinson [1,2,3]**, **Amandine Bodet [4]**, **Anne-Charlotte Monteau [5]**, **Sigfredo Fuentes [1]**, **Geoffrey Scollary [6]**, **Trevor Smith [6]** and **Kate S. Howell [1,*]**

1    School of Agriculture and Food, Faculty of Veterinary and Agricultural Sciences. The University of Melbourne, Building 142 Royal Parade, Parkville, Melbourne, VIC 3010, Australia
2    Montpellier SupAgro, 2 Place Pierre Viala, 34060 Montpellier, France
3    Hochschule Geisenheim University, Von-Lade-Straße 1, 65366 Geisenheim, Germany
4    Institut National Polytechnique - École Nationale Supérieure Agronomique de Toulouse, 31326 Castanet, Tolosan, France
5    Domaine Chandon, 727 Maroondah Hwy, Coldstream, VIC 3770, Australia
6    ARC Centre of Excellence in Exciton Science, School of Chemistry, The University of Melbourne, Parkville, Melbourne, VIC 3010, Australia
*    Correspondence: khowell@unimelb.edu.au

**Abstract:** The appearance of bubbles and foam can influence the likeability of a wine even before its consumption. Since foams are essential to visual and taste attributes of sparkling wines, it is of great importance to understand which compounds affect bubbles and foam characteristics. The aim of this work was to investigate the effect of interactions among proteins, amino acids, and phenols on the characteristics of foam in sparkling wines by using synchronous fluorescence spectroscopy techniques. Results have shown that several compounds present in sparkling wines influence foam quality differently, and importantly, highlighted how the interaction of those compounds might result in different effects on foam parameters. Amongst the results, mannoproteins were found to be most likely to promote foam and collar stability, while phenols were likely to increase the small bubbles and collar height in the foam matrix. In summary, this work contributes to a better understanding of the effect of wine compounds on foam quality as well as the effect of the interactions between those compounds.

**Keywords:** fizz; bubbles; protein; yeast invertase; foamability; wine quality

---

## 1. Introduction

The visual appearance of a wine is the attribute that provides the first impression for the drinker [1,2]. In the case of sparkling wines, since the vast majority result from secondary fermentation of white wines, bubbles and foams are of even greater importance [3]. The appearance of bubbles and foam are judged subjectively by wine appreciators [4,5], which might set the likeability of the wine even before its consumption. Furthermore, bubbles' size and distribution influence foam texture, thus influencing the sensory experience of the wine consumer [6,7]. Therefore, since foams alter visual, aroma and taste attributes of sparkling wines, it is essential to know the compounds that affect bubble and foam quality. Understanding the complex mechanism of foam and bubble formation, stabilization, and how it relates to overall wine quality will provide tools for wine producers and researchers to optimize wines with desired and improved qualities.

A thorough explanation of sparkling wine production methods and the compounds associated to the formation and stabilization of foam in sparkling wines has been recently presented by Kempt et al. (2018) [8]. Briefly, sparkling wines obtain the carbon dioxide by either natural fermentation, or via addition. Wines that are produced by *méthode traditionnelle* spend further time ageing on lees. Lees is a general name given to dead yeast cells that settle on the bottom of the vessel where the wine has been aging [9]. Wine aging duration is regulated by specific laws in countries such as France and Spain. Although, in Australia, there is no detailed laws regulating wine bottle aging as in Europe, the wines are aged for at least nine months on lees to be classified as *méthode traditionnelle*. Approximately 2–4 months after the second fermentation is finished, it is believed some compounds originated from yeast cytoplasm and cell walls are released into the wine matrix—this process is known as autolysis [9]. Nevertheless, our research team has been focusing particularly on the influence of proteins, amino acids and phenolic compounds on foam characteristics of sparkling wines, thus, emphasis will be given to those compounds.

The effect of proteins on foam properties of sparkling wines has been explored in the last 20 years by several authors [10–20]. The studies agree that proteins are the leading wine compounds influencing foam quality [8]. However, the effect on foam formation and foam stability is not clear, most likely due to different terminologies and methodologies used to assess foam characteristics. Since there is no official method to assess foam in sparkling wines objectively, different research groups have been using different methods to study foam characteristics, as well as different nomenclature, which makes it harder to compare the effect of chemical compounds on foam quality. Nonetheless, to be able to form a foam, proteins need to be rapidly adsorbed and unfolded at the gas/liquid interface, whilst to promote stability, it is necessary to create a robust and flexible film able to reduce gas permeability and bubble coalescence [21]. Proteins that are flexible, able to expose more hydrophobic residues and reduce the average molecular mass are good candidates to promote foam formation [21]. On the other hand, proteins that resist mechanical deformation and can form intermolecular cross-linking are good candidates to promote stability [21]. During the second fermentation of sparkling wines produced by using the *méthode traditionnelle*, amino acids, polysaccharides, peptides and proteins are released into the wine matrix [21]. Glycoproteins, such as plant derived, mannoproteins and yeast invertase are found in still white wines (Chardonnay) [15] and also released by yeast cells during the second fermentation [9]. Glycosilated plant derived proteins are highly hydrophobic, whilst mannoproteins found in wines vary in molecular mass from 53–560 kDa; and have a low ratio of protein/carbohydrate content, with mannose being the main carbohydrate component [22]. Vacuolar invertase originating from grapes, and yeast invertase resulting from wine fermentation are hydrophobic and the most abundant proteins found in wines [11,23,24]. Since the majority of proteins found in sparkling wines are hydrophobic [25], it could be expected that most proteins found in wines are more likely to be involved in foam formation rather than foam stability. However, in solution, proteins might interact with other compounds through electrostatic and hydrophobic forces, hydrogen bonds and covalent linkages, and might bind to one another resulting in no free molecules [21]. Consequently, the protein's properties are modified, and so, the resulting effect on foamability and stability is unclear. Furthermore, the formation and stability of foams in sparkling wines has been shown to result from complex interactions from the different compounds found in the wine matrix, such as monosaccharides [10], protein-protein interactions [15], polypeptides [26], rather than a single family of compounds.

The available literature on the effect of amino acids on foam quality is limited and inconsistent. Some studies have found positive correlation between the amino acids and foam characteristic [27,28], while others have found no correlation [29]. Our research group has recently found the influence of amino acids on foam quality was dependent on the production method of sparkling wines [20]. In this study, we have found that an increase of asparagine content would result in decreased foam stability, for the wines elaborated by the *méthode traditionnelle*. Additionally, we have found that wines elaborated by the carbonated method had increased foam stability with an increase in protein content, although those wines had lower protein content than those which were submitted to a second

fermentation. Moreover, carbonated wines had lower content of the amino acid asparagine. Thus, asparagine could have a negative effect on the foam of sparkling wines, especially in those wines which undergo a second fermentation. Therefore, we had hypothesized that an interaction of asparagine and wine proteins could offset the positive relationship of proteins on foam quality found for the carbonated wines.

Phenolic compounds have an important role in wine quality, being not only involved in browning and bitter tastes of white wines, but also involved in colour, astringency, solubility and volatility of aroma compounds. They include all compounds with hydroxyl groups attached to aromatic rings [23]. Polyphenols have multiple phenol rings within a single structure, such as epicatechin [23]. They are subdivided as flavonoids and non-flavonoids. Flavonoids have a particular C6–C3–C6 three-ring structure with a central oxygen-containing ring [23]. Non-flavonoids are known as hydroxycinnamates and are the major class of phenols found in wines [23]. Although total phenolics have been shown to negatively correlate to quality ratings of sparkling wines [24], the influence of phenols in foam formation and stabilization of sparkling wines is largely unknown. It has been suggested that phenols might influence positively the foam of sparkling wines produced using red grapes, such as Pinot noir [25]. On the other hand, although the interactions between phenolic compounds and proteins make it possible to eliminate the proteins responsible for haze formation [26], this could be detrimental to foam quality if it impairs the probability of the protein being able to act as a foam formation agent or stabilizer. The extent of proteins-polyphenols interactions is dependent on molecular size, number, and disposition of phenolic nuclei, conformational flexibility and water solubility of a specific polyphenol [27].

Polyphenols and proteins form soluble and insoluble complexes [30]. The mechanisms involved in reversible/non-covalent forces are hydrogen bonding [31,32], Van der Waals forces [33], hydrophobic bonding [34,35], non-polar hydrophobic interactions [36], hydrophobic and hydrophilic interactions [37]. On the other hand, non-disulphide covalent linkages [38,39] and cross-linking are involved in covalent/irreversible polyphenol-protein interactions. The nature of polyphenol-protein interaction is dependent on the molecular size, flexibility and water solubility of polyphenols [40], as well as the ratio of polyphenols: proteins [41]. When this ratio is low, is more likely that proteins and polyphenols will form large complexes and further precipitation [41]. This is because proteins have limited sites where polyphenols are able to link, thus, for a low polyphenol: protein ratio, the polyphenols are able to bind to more than one protein and form bridges, which then results in formation of large complexes and further precipitation [41]. However, if the ratio is high, precipitation is less likely to occur, since the formation of large complexes is not favoured.

Fluorescence spectroscopy is a rapid method that can support the identification of compounds [42,43] and assessment of interactions in wine [44–46] and, more specifically, interactions between proteins and other compounds [47,48]. The aromatic amino acids tryptophan, tyrosine and phenylalanine are fluorophores found naturally in most proteins, with tryptophan being one of the most prevalent [49]. The fluorescence of tryptophan is highly dependent on the environment; thus, the study of the fluorescence pattern of tryptophan can assist in our understanding of changes in protein conformation and the interactions of proteins with other compounds [49].

Therefore, based on the available literature and results found in a survey study of the foaming parameters associated to sparkling wines elaborated by different production methods [20] several hypotheses were formulated: (i) proteins and amino acids influence foam properties, such as foam stability; (ii) compounds interact with proteins resulting in a positive or negative effect on foam quality; and (iii) proteins (principally those originating from yeast) and amino acids increase the average foam lifetime, $L_f$.

To test these hypotheses, several compounds were added to a sparkling wine in order to isolate the effect on foam quality: (i) yeast invertase, bovine serum albumin, asparagine, tryptophan; (ii) yeast invertase + gallic acid, yeast invertase + asparagine; (iii) yeast invertase. Then, to confirm whether compounds present in sparkling wines interact with proteins, fluorescence spectroscopic analysis

was used. Finally, analysis of several foam parameters was performed, and the results obtained were assessed by statistical analysis.

## 2. Materials and Methods

### 2.1. Wine Material

Sparkling wine samples were supplied by Domain Chandon, Yarra Valley, Victoria, Australia. The samples were composed of a sparkling white wine produced by following the *méthode traditionnelle*, where a second fermentation is realized in the bottle and subsequently, the wine is subjected to a lees aging process [8]. The base wine was composed of Chardonnay (50%), Pinot Noir, Pinot Meunier and Pinot Gris wines. The grapes originated from the Yarra Valley (VIC, Australia; 91%) and the remainder from the vineyards located in Strathbogie Ranges, King Valley and Goulburn Valley, all located in Victoria, Australia. The musts of each grape variety are fermented separately, and the final blend was achieved after tastings and chemical analysis, in order to meet the desired quality standards. The final blend of base wine, being non-vintage, was allowed by Australian regulations to have a proportion of wines from previous vintages (reserve wines). Thus, the final blend was elaborated with 60% of 2016 wines and 40% of reserve wines, mainly from the previous vintage of 2015. The yeast used for the second fermentation was the Lalvin EC-1118® (Lallemand, Edwardstown, South Australia, Australia) and the fermentation was conducted at 17 °C. Since the aim of this study was to uncover the effect of specific compounds on foam quality in sparkling wines, we have selected a wine that has been aged for three months on lees. Thus, a wine in its initial stages of autolysis might reduce the complexity of analysis that would result from having autolytic compounds in abundance, and so, makes it easier to isolate the effects of the compounds here selected and investigated. It is worth noting that those samples were destined for research purposes only, and were not meant to be commercialized or consumed.

### 2.2. Basic Wine Composition for the Control Sample

The measurements were made in triplicates and results were recorded as mean ± standard deviation. *Soluble protein content* was measured following the methodology presented by Condé and colleagues [20]. *Alcohol by volume (%)* was determined by using Wine Alcolyzer (Anton Paar). Total phenolics and hydroxycinnamic acids, glucose, fructose, pH and organic acids were determined according to the methods given by Iland and colleagues [50].

### 2.3. Wine Samples

Chemical compounds (9) were added to the wine samples during the disgorging process, at a concentration of 100 mg/L. The samples and abbreviations are summarized in Table 1. In order to better assess the effect of specific compounds on foam parameters, the same concentration for all the compounds here analysed was selected. The concentration of 150 mg/L was found to be the median for protein content in a set of 28 wines, and average of slightly higher than 150 mg/L. The average for each of the amino acids analysed was lower than 100 mg/L. Thus, the decision to compare 100 mg/L for all the compounds was taken as a middle term between the content found for the proteins and amino acids in our previous study [20].

**Table 1.** Summary of samples, chemical additions and corresponding abbreviations.

| Sample | Concentration Added | Abbreviation |
| --- | --- | --- |
| Control | n.a. | Ctrl |
| Alcohol | 0.1% ABV | Alc |
| Yeast Invertase | 100 mg/L | INV |
| Bovine Serum Albumin | 100 mg/L | BSA |
| Manonoprotein | 100 mg/L | MAN |
| Gallic Acid | 100 mg/L | Gall |
| Asparagine | 100 mg/L | Asn |
| Tryptophan | 100mg/L | Trp |
| Yeast Invertase + Asparagine | 100 mg/L + 100 mg/L | INVAsn |
| Yeast Invertase + Gallic Acid | 100 mg/L + 100 mg/L | INVGall |

n.a. not applicable.

### 2.4. Chemical Additions

Albumin from bovine serum (≥96%) (Sigma-Aldrich Pty. Ltd., Saint Louis, MO, USA), Yeast invertase from baker's yeast (≥300 units/mg solid [51]) (*S. cerevisiae*, Sigma-Aldrich Pty. Ltd., Saint Louis, MO, USA); mannoproteins (Mannolees™Lallemand, Edwardstown, South Australia, Australia, commercial product composed from specific yeast mannoproteins); DL-asparagine (≥98%) (Thermo Fisher Scientific, Heysham, Morecambe, UK); L-tryptophan (≥98%) (Sigma-Aldrich Pty. Ltd., Saint Louis, MO, USA), gallic acid monohydrate (≥98%) (Sigma-Aldrich Pty. Ltd., Saint Louis, MO, USA).

### 2.5. Determination of Foam Parameters

Foam parameters were obtained by analysing two bottles of each treatment. Each bottle was analysed, using a robotic pourer, according to the methodology previously described by Condé and colleagues [52]. The wines were at room temperature (18 °C) before pouring. The foam parameters quantified included: foam volume ($V_f$); foam time ($F_t$); average foam lifetime ($L_f$); collar height ($h$); foam velocity ($F_v$); drainability ($D_r$); percentage of wine in the foam ($W_f$); collar initial height ($h_c$); foam expansion ($E$); and small bubbles ($S_b$). The foam parameters quantified were the average of triplicate measurements per bottle.

### 2.6. Fluorescence Spectroscopy

Fluorescence spectroscopy was applied to facilitate the identification of possible compounds that could be further related to foam quality. Additionally, to investigate protein interactions, fluorescence emission measurements recorded with an excitation wavelength ($\lambda_{ex}$) of 278 nm were further examined. Tryptophan is excited around 280 nm and emits around 350 nm in proteins [53]. Furthermore, the fluorescence intensity (FI) was calculated by integration of the area under the curve in an emission range $\lambda_{em}$ 300–400 nm, following by $\lambda_{ex}$ at 278 nm, for each sample.

The fluorescence excitation/emission matrices spectral data were obtained from each sample using a Varian Cary Eclipse fluorescence spectrophotometer operated in a synchronous scan mode using the front faced geometry (which has been shown to minimize issues associated with reabsorbance, inner filtering and scattering [54,55]) with the sample in a triangular quartz cuvette (path length 10 mm) mounted on a rotational mount to provide a 55° angle of the front face relative to the direction of the incident excitation beam. The use of the synchronous scanning mode reduces scattering originated from regions where $\lambda_{em} \sim \lambda_{ex}$ [56]. The excitation range was set to 205–405 nm, and the corresponding synchronized emission was set to start at delta 10 nm (215 nm), with delta increments of 5 nm and the delta stop set to 200 nm (650 nm). The detector was operated at 800 volts to ensure low intensity fluorophores were recorded. All spectra were recorded at room temperature.

## 2.7. Parallel Factor Analysis (Parafac)

Each two-dimensional excitation/emission matrix (EEM) obtained from the fluorescence from each sample was overlaid into a three-dimensional array of data (X) with dimensions 'sample x emission x excitation'. The X matrix was further smoothed and analysed using the Matlab Toolbox drEEM [57]. The selection of the number of components was based on the core consistency and percentage of explanation of the data. Subsequently, the excitation/emission ($\lambda_{ex}/\lambda_{em}$) patterns identified by Parafac analysis were further explored by principal component analysis in order to uncover chemical compounds that could possibly influence foam quality. Figures were obtained by using Minitab®17.1.0 (Minitab Incorporated., Pennsylvania State University, State College, PA, USA) and Matlab®2017b (MathWorks® Incorporated, Natick, MA, USA).

## 2.8. Statistical Analysis

Statistical data analyses included the general linear model (Glm) and analysis of means (ANOM) were performed using SAS 9.3 (SAS Institute Incorporated., Cary, NC, USA) and Minitab® 17.1.0 (Minitab Inc., Pennsylvania State University, State College, PA, USA), respectively. Glm was used to assess whether there were any differences between each treatment and the control, for each parameter. ANOM provides visualization of the comparison for each group means against the overall mean, for each parameter.

## 3. Results

### 3.1. Wine Composition

The wine was found to contain residual sugar (glucose + fructose) at 0.5 g/L, malic acid of 2.4 g/L, acetic acid was 0.21 g/L and the wine pH 3.22. Titratable acidity measured as equivalent to tartaric acid at titration to pH 8.2 was 5.8 g/L. These measures were within specification for the commercial production of sparkling wine. Additionally, the soluble protein content was 23.4 ± 2.8 mg/L, the alcohol by volume measured as 12.1± 0.1% *v/v*, total phenolics at 1.8 ± 0.1 a.u. and hydroxycinnamic acids at 1.0 ± 0.1 a.u.

### 3.2. Effect of the Chemical Additions on Foam Parameters

Several foam parameters were analysed using an automated pourer with image analysis. The parameters foam time ($F_t$), average foam lifetime ($L_f$), collar initial height ($h_c$), percentage of wine in the foam ($W_f$), foam expansion ($E$), did not show significant differences when compared to the control. The parameter foam volume ($V_f$) was significantly decreased by addition of INV, BSA, Trp, INVAsn and INVGall to the wines. The parameter foam velocity ($F_v$) and drainability ($D_r$) were also significantly decreased by addition of the amino acid Trp. Additionally, the parameter collar height ($h$) was significantly increased by MAN addition, and the measure of small bubbles ($S_b$) was significantly decreased in the wines with the addition of Asn. A summary of the parameters significantly different from the control sample, together with the direction of the effect on the parameter, is shown in Table 2.

Furthermore, the average mean per treatment (sample), for each foam parameter was analysed and compared between the group means, and significant differences ($\alpha$ = 0.05) were found for the parameters foam volume ($V_f$), foam velocity ($F_v$), average foam lifetime ($L_f$), collar height ($h$), drainability ($D_r$), foam expansion ($E$), and small bubbles ($S_b$) (Figure 1). Figure 1 shows the different compounds assessed were found to influence differently those foam parameters. BSA was found to significantly increase $L_f$ (Figure 1). Asn and Gall significantly increased $V_f$, while Inv, InvGall and Trp decreased $V_f$, Asn was found to increase $F_v$ and decrease $S_b$; MAN was found to increase $h$ and $E$; Gall increased $D_r$ and $S_b$ (Figure 1).

**Table 2.** Foam parameters analysed showing significant increase (↑) or decrease (↓).

| Sample | Foam Volume ($V_f$) | Foam Velocity ($F_v$) | Drainability ($D_r$) | Collar Height ($h$) | Small Bubbles Ratio ($S_b$) |
|---|---|---|---|---|---|
| Control | n.a. | n.a. | n.a. | n.a. | n.a. |
| Alcohol | n.s. | n.s. | n.s. | n.s. | n.s. |
| Yeast Invertase | ↓ | n.s. | n.s. | n.s. | n.s. |
| Bovine Serum albumin | ↓ | n.s. | n.s. | n.s. | n.s. |
| Mannoprotein | n.s. | n.s. | n.s. | ↑ | n.s. |
| Gallic Acid | n.s. | n.s. | n.s. | n.s. | n.s. |
| Asparagine | n.s. | n.s. | n.s. | n.s. | ↓ |
| Tryptophan | ↓ | ↓ | ↓ | n.s. | n.s. |
| Yeast Invertase + asparagine | ↓ | n.s. | n.s. | n.s. | n.s. |

n.a. not applicable. n.s. not significantly different (*p*-value > 0.05).

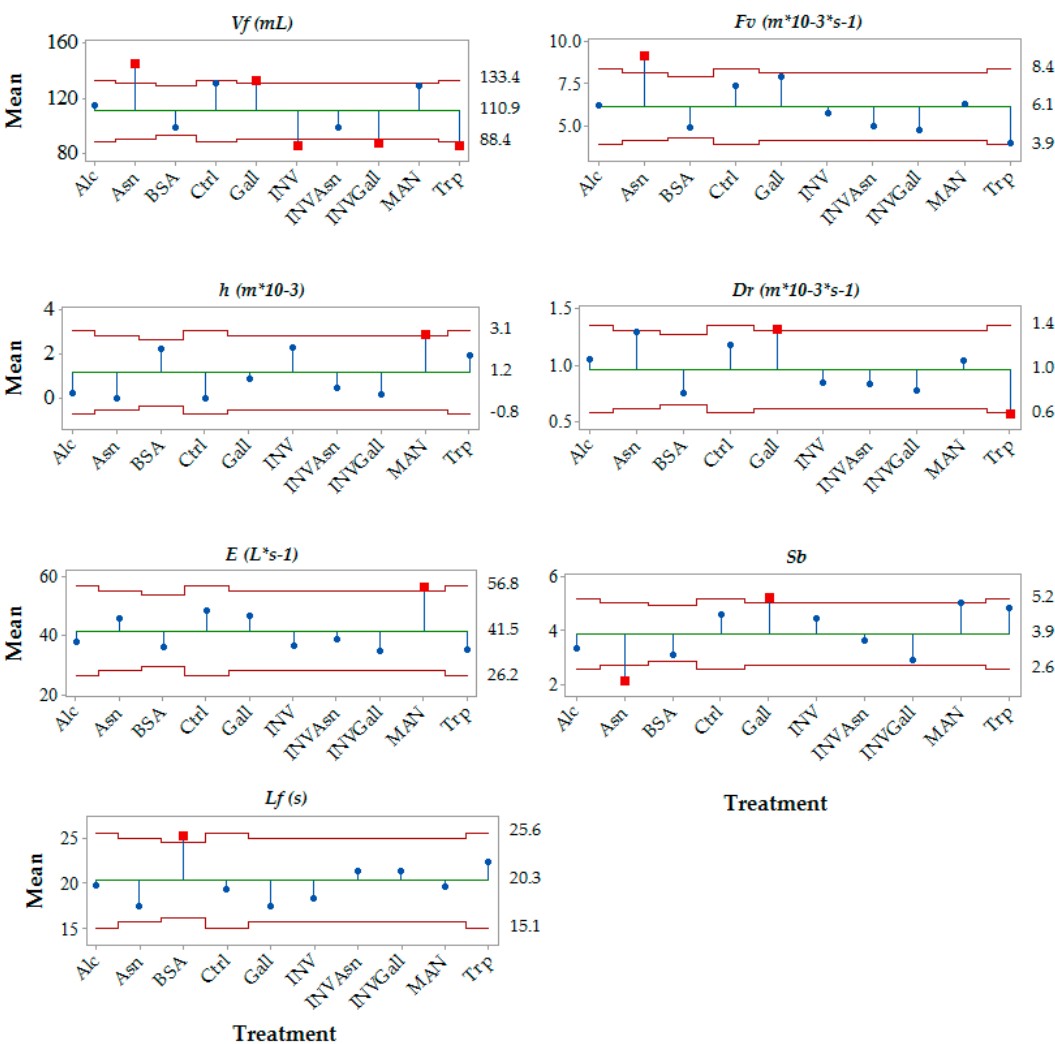

**Figure 1.** Foam parameters for wine treatments with added chemical compounds. The left and right 'y' axis, of each parameter, represents the values, in same units as found in the parentheses, next to each parameter. The average mean within a parameter, considering all samples, is represented by the green horizontal line; the mean value per sample is represented by the blue dot and when significantly different from the group means ($\alpha = 0.05$), the sample mean is highlighted as a small red square situated outside the limits. The red line represents the 95% interval limit. $V_f$, foam volume; $F_v$, foam velocity; $h$, collar height; $D_r$, drainability; $E$, foam expansion; $S_b$, small bubbles; $L_f$, average foam lifetime.

### 3.3. Spectrofluorescence and Parafac Analysis

Contour plots were generated to assist visualization of the emission/excitation wavelengths and intensities for each sample, as well as identify possible interactions among the compounds. Figure 2 suggests a slight decrease in fluorescence emission for the samples Gall and INVGall, when compared to Ctrl, and an increase of fluorescence emission for the remaining samples. Hence, the preliminary analysis of the raw data suggested possible molecular interactions between the components present in the wine matrix might be responsible for the changes in fluorescence intensity.

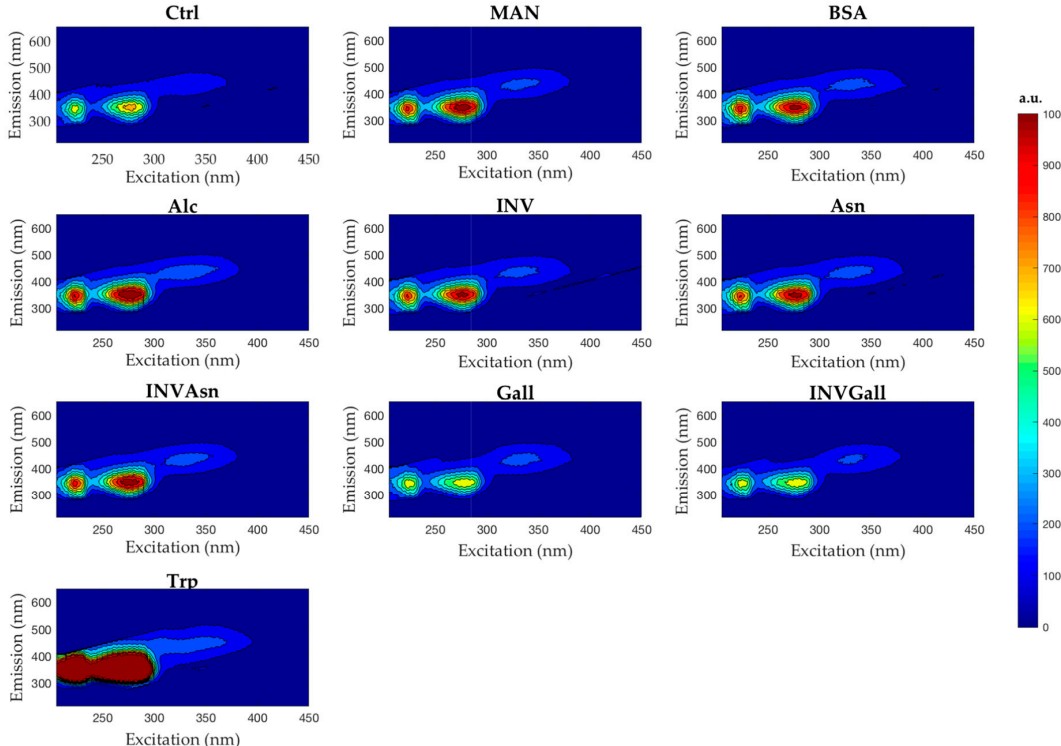

**Figure 2.** Fluorescence excitation and emission contour plots of the experimental wines. Ctrl, control; MAN, mannoprotein; BSA, bovine serum albumin; Alc, alcohol; INV, yeast invertase; Asn, asparagine; INVAsn, yeast invertase + asparagine; Gall, gallic acid; INVGall, yeast invertase + gallic acid, Trp, tryptophan.

The wines with added gallic acid, Gall, showed lower fluorescence intensities than those wines with added invertase, (INV). When these compounds were added together to a wine (INVGall), which showed a slight decrease of fluorescence when compared to Gall, and a considerable decrease of intensity when compared to INV (Figure 2). The reduction in fluorescence (quenching) caused by the addition of Gall could be due to molecular interactions between gallic acid and compounds present in the sparkling wine. The interactions between Gall and INV, may also be affected by other interactions between Gall and other molecules present in the control sample.

In order to better understand the interactions between proteins and other compounds present in the wine, the emission following excitation at 278 nm was examined and the results are shown in Table 3. The results showed a decrease in fluorescence intensity (FI) for the wines with added Gall and INVGall and an increase in FI for the remaining samples when compared to the Ctrl wine. A small decrease in $\lambda_{em}$ was observed for all samples, suggesting that the tryptophan present in the proteins of the control interacted with the compounds added and thus quenched some emissions. When observing the emission wavelength and FI for Gall and INVGall, it is noted that the proteins have an excellent affinity to phenols, most likely causing changes in protein conformation that results in fluorescence quenching and a small tryptophan blue shift [53].

The Parafac model selected had two components showing 65.8% core consistency and could explain 95.8% of the data variation. The $\lambda_{ex}/\lambda_{em}$ pattern obtained by the Parafac model is shown in Figure 3. The results obtained by the first component are defined by two regions of $\lambda_{ex}/\lambda_{em}$ = 222–249/330–360 nm and $\lambda_{ex}/\lambda_{em}$ = 274–285/350 nm (Figure 3a,b); the second component showed one region with maximum $\lambda_{ex}/\lambda_{em}$ = 326-345/435 nm (Figure 3c,d). The Parafac scores, when compared to foam parameters, did not show any clear correlations (data not shown), thus, principal component analysis (PCA) was applied. Three principal components were required, corresponding to three regions (emissions at 245 nm, 350 nm and 435 nm). The results for each model and the scores related for each variable for each principal component are described in Table S1 (supplementary data). The PCA did not show any significant relationship for the region of $\lambda_{ex}/\lambda_{em}$ = 222–249/330–360 nm. However, for the $\lambda_{ex}/\lambda_{em}$ = 274–285/350 nm region, a positive relationship with collar height ($h$) and a negative relationship with drainability ($D_r$) was found. In addition, a positive relationship was found with collar initial height ($h_c$) for the second component (region $\lambda_{ex}/\lambda_{em}$ = 326–345/435 nm).

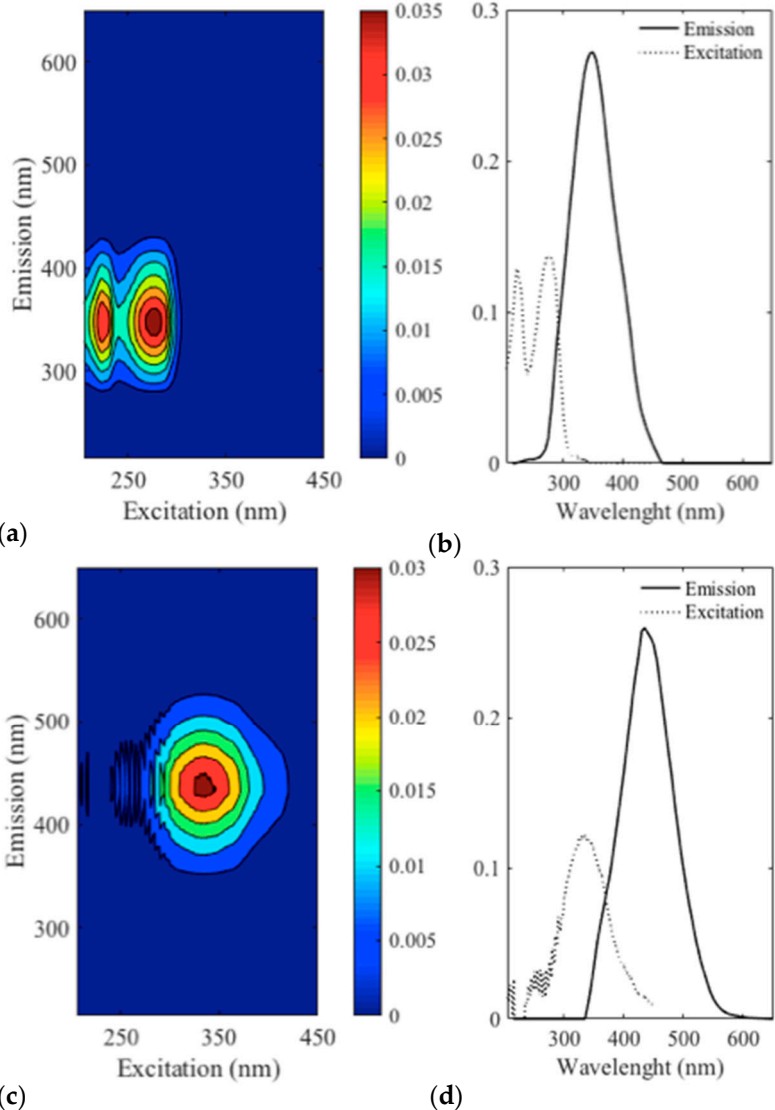

**Figure 3.** Emission and excitation emission matrix and Parafac fingerprints obtained from the experimental wines (**a**) contour plot for component 1 and (**b**) respective loading plot; (**c**) contour plot for component 2 and (**d**) respective loading plot.

**Table 3.** Emission wavelength and the corresponding fluorescence intensity following $\lambda_{ex}$ 278 nm.

| Sample | $\lambda_{em}$ | FI |
|---|---|---|
| Control | 350 | 52,222 |
| Alcohol | 335 | 75,279 |
| Asparagine | 335 | 73,024 |
| Bovine Serum Albumin | 340 | 72,225 |
| Gallic Acid | 340 | 46,462 |
| Yeast Invertase | 340 | 70,930 |
| Yeast Invertase + Asparagine | 335 | 73,377 |
| Yeast Invertase + Gallic Acid | 345 | 45,953 |
| Mannoprotein | 335 | 73,257 |
| Tryptophan | 305 | 86,562 |

## 4. Discussion

The purpose of this study was to investigate the effect of several compounds, and interactions amongst proteins, amino acids, and polyphenols on the characteristics of foam in sparkling wines by exploring several hypotheses.

### 4.1. Identification of Compounds Related to Foam Quality

The fingerprint for the samples analysed by fluorescence spectroscopy was presented by the principal components (Figure 3a–d). The figure has indicated the presence of several compounds, such as those originated from yeast/bacteria, polyphenols and proteins that were further associated to foam parameters, which is discussed in the following session.

The left region of the first component is characterized by a peak of short excitation and short emission wavelength (Figure 3a,b), suggesting it is related to low molecular mass, simple aromatic compounds. Aromatic amino acids and nucleic acids (AAA + NA) are excited at 250–260 nm and emit fluorescence in the range of 331–336 nm [58,59]. Tryptophan residues found in the AAA + NA are the compounds that contribute mostly to the fluorescence, although its quantum yields are 100 times lower than the quantum yield of tryptophan alone [60]. Hence, component 1 could indicate the presence of AAA + NA, perhaps originating from *Lactobacillus* and/or yeast [58,59]. The pattern expressed by the second region of the first component $\lambda_{ex}/\lambda_{em}$ = 285/350–363 nm is more likely to have resulted from the fluorophores from different polyphenols and proteins. Polyphenols are excited in the range of 260–330 nm and show emission in the range 310–442 nm [61]. Catechin, epicatechin, epigallocatechin and procyanidin exhibit a pattern $\lambda_{ex}/\lambda_{em}$ = 280–290/310–320 and gallic acid $\lambda_{ex}/\lambda_{em}$ = 278–280/320–366. Nevertheless, proteins also have similar fluorescence patterns ($\lambda_{ex}/\lambda_{em}$ = 279–295/300–350) [53]. The compounds responsible for fluorescence in this region were found to be positively related to collar height (*h*), and negatively related to drainability (*$D_r$*). The addition of mannoproteins was found to increase *h* (Figure 1), thus, they are the most probable compounds related to the findings of the PCA regarding the first component of the Parafac analysis. Consequently, we can determine that proteins present in the wine matrix do influence foam stability by decreasing the drainability (Figure 1) and maintaining the presence of a collar for the duration of 300 s (definition of *h* [52]).

The region represented by the second component ($\lambda_{ex}/\lambda_{em}$ = 326–345/435 nm) likely results from the fluorescence of the hydroxycinnamic acids (HCAs) present in the sparkling wines. HCAs have absorption maxima around 325 nm and maximum fluorescence emission at 440 nm [62] and are present naturally in white wines [63] originating from grape pulp [64]. However, they are found in the form of tartaric acid esters highly susceptible to hydrolysis [64].

An unexpected result observed was the significant influence of polyphenols and hydroxycinnamates on foam parameters. Polyphenols, most likely non-flavonoids, appear to affect positively the foam characteristics, most importantly, the measure of small bubbles (*$S_b$*) (Figure 1) and collar initial height (*$h_c$*) (Table S1). This is the first study to report a wine matrix component that is

able to increase the parameter $S_b$. It is desirable to have a high number of small bubbles in sparkling wines, as they are highly sought after and visual assessment of these bubbles enhances perceived quality. However, when existing simultaneously with INV, phenols might negatively effect bubble size as observed in Figure 1. A recent study has found important correlations between foamability and stability, and anthocyanins, but did not find any influence of hydroxycinnamic acids (HCAs) on foam quality [28]. The discrepancy in the observed results is likely due to the different methodology applied to measure the foam parameters.

*4.2. Hypotheses*

Previous investigations on the effect of proteins and amino acids in foam parameters [20] has raised several hypotheses. This study has investigated these specific hypotheses, which are discussed in detail below.

(i)　Proteins and amino acids influence foam properties, such as foam stability

Our hypothesis testing (i) has shown that proteins, such as yeast invertase, bovine serum albumin and amino acids, such as asparagine and tryptophan significantly affect foam properties, principally those associated to foam stability ($D_r$, $F_v$, $L_f$). Amino acids might promote foamability but might have a negative effect on foam stability (Figure 1). Also, different proteins have different compositions resulting in specific interactions to wine compounds, consequently affecting foam quality differently. For instance, BSA forms hydrogen bonds/hydrophobic interactions to proanthocyanidins and catechins originated from grapes and white wines, resulting in an increase of hydrophobicity of the molecules [65]. On the other hand, the different composition and conformation of glycosylated proteins increase the possibility of binding to phenolic compounds and decrease the potential to form and stabilize foams.

Although it might be anticipated that yeast invertase (INV) and mannoproteins (MAN) would show similar effects on foam parameters, it is worth noting that, while INV had a high purity, MAN was a commercial product which contains other compounds that might impact how the proteins interact with other wine chemicals. The different effect of INV and MAN on foam parameters is observed in Figure 1. For instance, apart from ($S_b$), MAN and INV had opposite effects for the foam parameters analysed—MAN showed positive effect while INV showed a negative effect. Mannoproteins are generally composed of 20% proteins and 80% D-mannose associated with D-glucose [66] and have been found to have an affinity to flavonols [67]. Yeast invertase, like mannoproteins, has a lower ratio of proteins to glycosylated compounds (50% mannose/2–3% glucose) [51,68]. The association of MAN and INV with flavonols is likely to happen between the glycosylated moieties rather than to the protein side [68]. The negative effect of INV on foam parameters was more evident when Gall was also added to the sample (Figure 1).

We speculate that the significant influence of BSA on the average foam lifetime ($L_f$) (Figure 1) could be related to an increase in hydrophobicity in proteins (showing similar behaviour to BSA) caused by interactions to the phenolic compounds present in the wines, most likely proanthocyanidins, or is promoted by proteins which are hydrophobic. Hydrophobic compounds have more affinity to gas and other compounds in the bubble walls [25], resulting in a resistant viscoelastic film [11], thus promoting foam stability. On the other hand, the complex formed between the phenolic compounds and glycosylated proteins seems to be detrimental to foam quality.

(ii)　Compounds interact with proteins resulting in a positive or negative effect on foam quality

The interaction between proteins and other compounds was suggested by the contour plots (Figure 2) and become more evident when assessing the fluorescence intensity (FI) and emission wavelength shifts that occurred when $\lambda_{ex}$ is 278 nm. The FI increase observed for bovine serum albumin (BSA), yeast invertase (INV), yeast invertase + asparagine (INVAsn), mannoproteins (MAN), and tryptophan (Trp) (Table 3) is likely due to an increased concentration of tryptophan residues while the increase in FI seen for alcohol (Alc) and asparagine (Asn) (Table 3) is more likely to be

caused by structural changes undergone by the proteins present in the wine in the presence of those compounds, leading to exposure of buried tryptophan residues [69]. The fluorescence quenching observed in the presence of gallic acid (Gall) and yeast invertase + gallic acid (INVGall) (as shown by a decrease of fluorescence intensity observed in Figure 2 and Table 3) highlights the interaction between those compounds, and also, interactions to other compounds present in the wine matrix. Furthermore, the quenching of tryptophan emission is indicative of tryptophan location within a protein [70] and is also indicative of changes in protein conformation [53]. The tertiary structure of proteins has been found to change its conformation when binding to phenolic compounds [71]. Tryptophan (Trp) residues when fully exposed emit at 350 nm; partially exposed, at 340 nm; buried within a protein, but interacting to the surrounded environment, at 315–330 nm; and fully buried (within a protein), at 308 nm. The observed blue shift (Table 3) for the samples INV, MAN and BSA is probably the result of hydrogen bonds between –OH moieties in polyphenols and the $NH_2$, OH and SH groups in the proteins (yeast invertase, mannoproteins and bovine serum albumin) [71,72]. Additionally, the considerable blue shift observed for Trp highlights the high affinity of tryptophan to the compounds present in the wine sample—most likely polyphenols. Furthermore, it is worth noting that for the control sample, the emission was 350 nm (Table 3), suggesting the tryptophan residues are fully exposed, thus, the proteins present in the wine sample are not strongly interacting with the polyphenols present in the wine. However, when gallic acid is added to the samples, the blue shift is noticeable, as well as a decrease in the fluorescence intensity, which shows the high affinity of the proteins present in the control wine with gallic acid. In summary, the changes in the proteins' conformation due to interactions with polyphenols might impair the likelihood of the protein being adsorbed in the foam matrix. Therefore, foam stability and foam formation are negatively affected in finished sparkling wines, which are likely to have mannoproteins and/or yeast invertase present in the wine matrix.

The interactions between proteins and amino acids were also shown to impair foam quality (Figure 1). For instance, the foam promoting effect of Asn was neutralized by INV (Figure 1) and the positive impact of yeast proteins and phenols on bubbles size, observed by an increase of the number of small bubbles ratio as seen in Figure 2 (Gall and INV increased $S_b$), was counteracted when both compounds were added simultaneously—A decrease in $S_b$ was observed. Our work suggests that the high affinity of phenolic compounds to the wine proteins means that proteins are likely be found bound to compounds, resulting in a negative effect on foam quality, or a positive effect (such as found when BSA was added—discussed in the hypothesis above).

While the use of fluorescence spectroscopy was able to reveal interactions between polyphenols and proteins, the methodology used was not able to categorize the nature of those interactions. Further studies are being carried out to uncover the nature and other factors influencing those interactions.

(iii)　Proteins (principally those originated from yeast) and amino acids increase foam lifetime

The results did not support the hypothesis that yeast proteins and amino acids could increase the average foam lifetime ($L_f$) (iii). The parameter $L_f$ was found to be increased by the presence of BSA. When comparing $L_f$ obtained from the control and INV samples, it is observed that INV decreased $L_f$. The decrease in $L_f$ observed when adding yeast proteins might be due to strong interactions between INV and other compounds present in the wine sample, such as phenolic compounds, which resulted in the yeast proteins being unable to be absorbed in the film layer, and, consequently, unable to provide stability to the foam.

## 5. Conclusions

The results of the present study have shown that several compounds present in sparkling wines influence foam quality differently, as well as highlighting the importance of the interactions between these compounds and other wine components. Mannoproteins were found to be most likely to promote foam and collar stability, while phenols were likely to increase the number of small bubbles in the

foam matrix. Our work confirmed amino acids influence foam quality and showed how different proteins influence different foam parameters. Additionally, our study showed that polyphenols have high affinity to proteins present in sparkling wines, and this interaction might be positive or negative on foam characteristics. In summary, the interactions and the resulting effect on foam parameters in sparkling wines are extremely complex. However, the results discussed are limited by the concentration and specificity of amino acids, proteins and phenol here studied. Different concentrations, as well as other proteins and amino acids, might yield different results. Nonetheless, the findings here can be adequately explained, and the techniques here applied might assist us to better understand the consequences of those interactions and provide us with tools to be able to control and modify sparkling wine foam quality.

**Supplementary Materials:** The following are available online at http://www.mdpi.com/2306-5710/5/3/54/s1, Table S1: Variable scores for each principal component related to three principal component analysis ($\lambda_{em}$ 245 nm, $\lambda_{em}$ 350 nm, $\lambda_{em}$ 435 nm).

**Author Contributions:** Conceptualization: B.C., T.S., G.S. and K.S.H.; methodology: B.C., A.R., A.B. and A.-C.M.; software: B.C., T.S. and S.F.; validation: B.C., A.B., S.F.; formal analysis: B.C. and T.S.; investigation: B.C., A.R., A.B., A.-C.M. and K.S.H.; resources: T.S. and K.S.H.; data curation: B.C.; writing—original draft preparation, B.C.; writing—review and editing: B.C., A.R., A.B., A.-C.M., S.F., G.S., T.S. and K.S.H.; visualization: B.C. and A.R.; supervision: S.F., G.S., T.S. and K.S.H.; project administration: K.S.H.; funding acquisition: K.S.H.

**Funding:** This research was funded by WINE AUSTRALIA, grant number AGW Ph1508.

**Acknowledgments:** This study was supported by an Australian Government Research Training Program (RTP) Scholarship. We would like to thank Dan Buckle at Domaine Chandon, Yarra Valley, Australia, for providing the wine samples and enabling disgorgement of the wines.

**Conflicts of Interest:** The authors declare no conflict of interest.

## Abbreviations

| | |
|---|---|
| ($L_f$) | Average foam lifetime |
| ($h$) | Collar height |
| ($h_c$) | Collar initial height |
| ($D_r$) | Drainability |
| $\lambda_{ex}$ | Excitation wavelength |
| $\lambda_{em}$ | Emission wavelength |
| ($E$) | Foam expansion |
| ($F_t$) | Foam time |
| ($F_v$) | Foam velocity |
| ($V_f$) | Foam volume |
| ($W_f$) | Percentage of wine in the foam |
| ($S_b$) | Small bubbles |

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
