# Peer review of "Using Synchronous Fluorescence to Investigate Chemical Interactions Influencing Foam Characteristics in Sparkling Wines"

_beverages, doi:10.3390/beverages5030054_

Round 1

Reviewer 1 Report

The manuscript has been revised and in my opinion it can be accepted for publication.

Author Response

Thank you for these comments. 

Reviewer 2 Report

Review paper

 Using synchronous fluorescence to investigate compounds and interactions influencing foam characteristics in sparkling wines

Title is vague. What compounds? What interactions? Certain level of accuracy and precision is needed in the title of the manuscript. Please amend.

Abstract, line 24: “the ratio of small bubbles…” a ratio of a quotient between two figures, there is one figure missing here. 

Line 30. “In the case of sparkling wines, since the vast majority result from secondary fermentation of white  wines, bubbles and foams are of even greater importance.”

This is not accurate. There are around 600 million bottles of prosseco produced annually and these are sparkling wines produced without second fermentation in bottle. The authors need to be accurate and check the validity of their statements.

29-38: there are a lot of claims on this paragraph and none of them are backed up by published research. Authors need to add scientific peer reviewed references to back up each of these claims.

86-87: this is an oversimplification. Phenolics are not just involved in browning and bitterness, but also define color, astringency and even the solubility and volatility of aroma compounds. Please amend. 

94: be precise: it is not pinot noir, it is Pinot noir.

132-135: there is almost no details about the winemaking of the base wines. These need to be provided. Furthermore, there is no indications of the varieties used here, was it Pinot noir, chardonnay or both? Explain and address. 

134: is realized sounds awkward. Replace by “occurs” 

Lines 136 to 142 do not really belong to materials and methods. They belong to the discussion section, may be, or the introduction.

142 to 145: this is problematic and need to be addressed: the authors assumed that 3 months is a period that will allow analysis without the effect of autolysis. Partial autolysis may have occurred. Is there solid scientific evidence that points out towards 3 months as being a cut-off period for autolytic character? Please address.

152 to 158: there is no space between 100 and mg/L and it reads like 100mg/L, please add a space. Same in table 1. This is wrong.

166: why only 2 bottles of each treatment were analyzed? This would certainly decrease the statistical power of the analysis. Why not the standard 3 bottles per treatment statistical sampling protocol was followed?

173-175: again, this does not pertain to materials and methods, this pertains to the introduction or the discussion. The authors need to have a clear idea of what belongs to introduction, and what to materials and methods. Materials and methods should only detail the materials and methods, not the reasons or principles of the analyses carried out. 

Results section

General comments: the authors abused the use of acronyms. As a result, this manuscript is impossible to read and make sense of. The authors should remember that yes, they clearly know what all the various acronyms of foam characteristics are, but not the readership. Please address and correct. 

In table 2, what is the meaning of “n.a.”???

289: is the presence of lactobacillus likely here? Did the base wines undergo malolactic fermentation? This is not clear in the manuscript, please address.

307-309: authors should be careful when making inferences regarding polyphenols as these were measured according to Iland, and just as absorbance units. to be conclusive, various phenolic classes including hydroxycinnamic acids should have been measured by HPLC, which the authors did not do here. As such, the scope of their conclusions regarding the true effect of phenols on foam characteristics should be limited and confined to the method they used to determine phenols.

Author Response

Title is vague. What compounds? What interactions? Certain level of accuracy and precision is needed in the title of the manuscript. Please amend.

The title was modified to “Using synchronous fluorescence to investigate chemical interactions influencing foam characteristics in sparkling wines”

Abstract, line 24: “the ratio of small bubbles…” a ratio of a quotient between two figures, there is one figure missing here. 

This is a typo. We have removed ‘ratio of’ from the sentence.

Line 30. “In the case of sparkling wines, since the vast majority result from secondary fermentation of white  wines, bubbles and foams are of even greater importance.”

This is not accurate. There are around 600 million bottles of prosseco produced annually and these are sparkling wines produced without second fermentation in bottle. The authors need to be accurate and check the validity of their statements.

We understand that our expression of this sentence is misleading. We mean to say that when the wine is made using method tradionnelle, there is little room for manipulation of content by the wine producer as the wine is made in the same bottle it is sold in. We have clarified this sentence in the amended manuscript.

29-38: there are a lot of claims on this paragraph and none of them are backed up by published research. Authors need to add scientific peer reviewed references to back up each of these claims.

References were added to the section.

86-87: this is an oversimplification. Phenolics are not just involved in browning and bitterness, but also define color, astringency and even the solubility and volatility of aroma compounds. Please amend. 

The statement was amended.

94: be precise: it is not pinot noir, it is Pinot noir.

The statement was amended.

132-135: there is almost no details about the winemaking of the base wines. These need to be provided. Furthermore, there is no indications of the varieties used here, was it Pinot noir, chardonnay or both? Explain and address. 

We have added details of the winemaking process to the amended manuscript.

134: is realized sounds awkward. Replace by “occurs” 

The statement was modified as suggested.

Lines 136 to 142 do not really belong to materials and methods. They belong to the discussion section, may be, or the introduction.

The section was moved to introduction.

142 to 145: this is problematic and need to be addressed: the authors assumed that 3 months is a period that will allow analysis without the effect of autolysis. Partial autolysis may have occurred. Is there solid scientific evidence that points out towards 3 months as being a cut-off period for autolytic character? Please address.

According to Alexandre and Guilloux-Benatier (2006) yeast autolysis starts between 2-4 months – as stated in the manuscript. We chose a wine with a short period of time with autolysis- recognising of course that autolysis may have commenced- to reduce the autolytic features in the wine available to interact with our added compounds. We have clarified the sentence in the manuscript to make this explicit. Alexandre, H.; Guilloux-Benatier, M. Yeast autolysis in sparkling wine – a review. Aust J Grape Wine Res 2006, 12, 119-127. 

152 to 158: there is no space between 100 and mg/L and it reads like 100mg/L, please add a space. Same in table 1. This is wrong.

Spaces was added.

166: why only 2 bottles of each treatment were analyzed? This would certainly decrease the statistical power of the analysis. Why not the standard 3 bottles per treatment statistical sampling protocol was followed?

Each bottle was analyzed in triplicate, as explained in the manuscript. Statistical ‘power analysis’ (>0.8 and a=0.5) was performed in the initial stages of planning the experiment, and it was found that 2 bottles/treatment analyzed in triplicates were adequate.

173-175: again, this does not pertain to materials and methods, this pertains to the introduction or the discussion. The authors need to have a clear idea of what belongs to introduction, and what to materials and methods. Materials and methods should only detail the materials and methods, not the reasons or principles of the analyses carried out. 

The section was removed, as suggested there was redundancy with the introduction.

Results section

General comments: the authors abused the use of acronyms. As a result, this manuscript is impossible to read and make sense of. The authors should remember that yes, they clearly know what all the various acronyms of foam characteristics are, but not the readership. Please address and correct. 

We appreciate the complexity of our acronyms and have clarified this in the revised version of the manuscript. In particular, explaining these acronyms in full at the start of each section has been included. We believe this has improved readability and clarity of the manuscript.

In table 2, what is the meaning of “n.a.”???

Table 2 shows a summary of the parameters significantly different from the control sample, thus, the results are not applicable (n.a.) to the control sample itself. The acronym has been added to the table.

289: is the presence of lactobacillus likely here? Did the base wines undergo malolactic fermentation? This is not clear in the manuscript, please address.

We are not suggesting that Lactobacilli are present, rather that their biochemical products have been implanted in the wine during the winemaking process. It is entirely possible that the wines could have lactobacilli present without undergoing malolactic fermentation. We have clarified the production of the base wine in the revised manuscript.

307-309: authors should be careful when making inferences regarding polyphenols as these were measured according to Iland, and just as absorbance units. to be conclusive, various phenolic classes including hydroxycinnamic acids should have been measured by HPLC, which the authors did not do here. As such, the scope of their conclusions regarding the true effect of phenols on foam characteristics should be limited and confined to the method they used to determine phenols.

We appreciate the subtleties of the discussion the reviewer brings forward. We state in the conclusion that “the results here discussed are limited by the concentration and specificity of amino acids, proteins and phenol here studied”.

Reviewer 3 Report

This manuscript contains some useful information on the use of fluorescence to investigate the compounds which could influence the foam properties of sparkling wines. The methods employed were appropriate and the results appear adequate to provide an advance in the knowledge of this field.  However there are several weak points that need to be corrected in order to guarantee publication in Beverages journal. The comments regarding the above mentioned manuscript are the following:

Specific comments

line 102 Hydrophilic instead of hydropholic

Line 140-144: How the authors can justify that autolysis doesn't occur after three months of ageing on lees? It is a very important aspect that should be answered since the results might not be valid

Results: Line 207. Is there a dilution factor regarding total phenolics? 1.8 au is out of the range (0.1-1 au) for correct measurements.

Author Response

line 102 Hydrophilic instead of hydropholic

The word was corrected.

Line 140-144: How the authors can justify that autolysis doesn't occur after three months of ageing on lees? It is a very important aspect that should be answered since the results might not be valid.

The sparkling wine in initial stages of autolysis was chosen to make it easier to understand the isolated effect of the selected compounds. The authors have discussed the results found in a wine which was in its initial stages of autolysis to avoid compounds involved in autolysis interfering with the response of the added chemicals. The results and discussion are mainly focused on the effects of the compounds selected, and compounds identified by fluorescence spectroscopy.

Results: Line 207. Is there a dilution factor regarding total phenolics? 1.8 au is out of the range (0.1-1 au) for correct measurements.

Measurements with and without dilution were carried out and the results were similar.

This manuscript is a resubmission of an earlier submission. The following is a list of the peer review reports and author responses from that submission.

Round 1

Reviewer 1 Report

The article “Using synchronous fluorescence to investigate compounds and interactions influencing foam characteristics in sparkling wines” reports on the use of synchronous fluorescence technique to investigate how different compounds affect the foam characteristics of one sparkling wine, in the attempt to shed light on a topic that several other authors have worked on before. Using this technique is an element of novelty on these type of studies, and the article reports some interesting findings, but in my opinion the experimental design and the discussion of the findings present several issues.

My main criticisms are summarized below:

·  Part of the introduction needs to be rewritten as the background information presented by the authors is incomplete. In particular, several reviews and books are used as main sources for the background info, while the contributions of many authors (as Marchal, Ligier-Belair, Moreno-Arribas, … the list could go on and on) that worked a lot in this space have been ignored.

·  Materials and methods. The wine used has not been described, while, given that most of the conclusions drawn are referring to the wine composition (e.g. proteins, phenolics, aminoacids…), a thorough investigation on the key components needs to be given. This is required to avoid speculation during the interpretation of the results.

·   There are far too many acronyms used throughout the text. The reader can never remember their meaning and has to constantly try to refer back to the materials and methods section. Please, where possible, avoid the use of acronyms  

·   The discussion looks at interpreting the results of the synchronous fluorescence method to reasonably “guess” the presence of different classes of wine components as proteins, hydroxycinnamic acids etc. This is an interesting approach, but for the results to be more meaningful one needs to check, at least for some parameters, if the assumptions made are true, e.g. by quantifying with other methods the compounds discussed. Without this approach the discuss on the role of these compounds on foam parameters becomes really weak and speculative.

·  The conclusions section is too general, and by reading it one gets the wrong idea on the main findings of the paper. Please rewrite it

Below are some specific comments:

L22: incorrect use of 3rd person after result

L42: this sentence is debatable, and ref 2 cannot be identified as it is not reported correctly in the reference list

L49: French accents are missing in method traditionalle

L54-55: this sentence needs additional citations

L56: a citation is required when stating that the majority of sparkling wine proteins are hydrophobic. Please add one

L75: ref 9 is not readily available so cannot be checked, please provide a different citation

L78-79: the sentence that the literature lacks knowledge is inaccurate as there has been a lot of articles investigating this exact problem by groups in Spain, Italy, France, Chile, Australia etc. Please rephrase it and report correctly the background info required

L87-92: this paragraph indicates that the three theories mentioned were firstly drawn based on the results of ref 19, while this is clearly not the case as several other authors have been tackling these theories for almost 20 years now, starting with the works done in Champagne and continued in Cava. Please reword it, and consider all the relevant literature

L100-102: the knowledge of the wine chemical composition is essential for the interpretation of the data of this article and needs to be provided.

L107: why only three months on lees? Yeast autolysis is just starting with this limited timeframe.

L151: Murphy citation is not in the correct format

Table 2: there are too many abbreviations, to read the table one has to write on the top of the column what the acronyms for the parameters are and also what the samples ‘acronyms mean. Please at least write in full the treatments

Figure 1: same comment as before, it is annoying having to refer back to the mat and met section for understanding each graph shown. The legend could be at least listed in the caption of the figure

Table 3. please do not only use acronyms in the table, it makes it really difficult for the reader

195-196. maybe I am interpreting it wrong, but in table 3 I see a decrease in FI only for the MAN and Trp samples, while the text seems to comment different data, including INVGal that I do not find reported in table 3. This means that I do not know if I should believe to the following commentary (until L201) or to the numbers reported in Table 3.

L207: it should refer to figure 3c-d, not 4c-d

L212-215: here the authors refer to and describe a PCA that is however not shown in the article, with the exception of table S1 that is however not readily understandable by the reader. Can the PCA plot be presented as additional figure?

L228-230: the authors should better explain how the presence of several compounds as those mentioned here is supported by the data shown in figure 3

L237. Typo in Lactobacillus

L244: I understand that Mannose addition increased h, but how is figure 2 showing this? Perhaps you meant Figure 2?

L246: here the authors refer to the protein present in the wine matrix. Have these been measured? How do we know that the wine contains proteins? This information is critical and needs to be added

L250: same as previous comment, can the HCAs be measured to make these assumptions valid?

L254: as stated in the previous comments, you can only guess that you have polyphenols, but why not measuring them to avoid this hypothetical discussion?

L255: where do I have to look to see the polyphenols effect on foam characteristics?

L270-272: I think that this statement is not fully supported by the experimental findings. Only two amino acids were tested (Asn and Trp) and two proteins (BSA and INV), and I do not see enough evidence to support the claim made about amino acids and proteins. The claim can only be made regarding the aa and proteins tested, without generalizing.

L281: the ratio mannose/glucose should be reported as done for mannoproteins

L282-286: here it is justified why INV and MNA might behave differently, but the authors should also draw the attention of the readers on the differences actually observed for these two glycoproteins as shown in figure 1.

L299-301. It seems strange that a small alcohol change of 0.1% modify so greatly the proteins present in wine. The authors should comment on this more, possibly including a citation supporting this theory as it is not very convincing. Also, what was the starting alcohol concentration?

L301-303. This sentence is not sufficiently clear and convincing and therefore needs rewriting.

L307-309. Here the authors refer to proteins, but the discussion was about the sample added with tryptophan, not proteins that might or might not be present in the wine as this data was not presented. Please rewrite.

L324-327: this could be true, but if the wine proteins potentially present in the wines studies were to be bound to phenolic compounds they would most likely form insoluble complexes that would be removed at disgorging of the wines, so that the effect that the authors suggest that they might have on wine foamability would be true in theory but not relevant in practice. Please consider this comment to modify the discussion.

L329-334: the explanation given here is weak. Similarly to what said before, if the yeast proteins would interact with phenolics chances are that they would form insoluble complexes that would not be found in the finished wines.

L340. As said previously, the results here can only be associated with the effects of the amino acids tested without generalizing to all amino acids as the writing suggests.

L341. It would be better to be more specific rather than referring to” different proteins”  as this leads to a generalization of the findings not supported by the data

L342. I disagree, the results did not show that polyphenols (in general) have high affinity with sparkling wine proteins (in general) for the reasons mentioned earlier. Additionally, the reader does not know if the sparkling wine used had proteins and polyphenols, how much of these compounds are in the wine nor their identity. I think that it is not possible to extend the findings to make general considerations. Please carefully check the discussion and conclusion to avoid speculations

L335-347: the section is too general, and by reading it one gets the wrong idea on the main findings of the paper. Please rewrite it

Reviewer 2 Report

This paper investigates the influence of several compounds present in sparkling wines on foam quality, highlighting the importance of the interactions between these compounds and other wine components.

The proposed objectives were fulfilled and the paper is well written. The methods applied are adequate and the results are interesting. Despite the manuscript contains novel aspects, it presents some criticisms that make it unacceptable in its current form for publication in Beverages. Thus, I recommend publishing the paper after taking into account the following revision:

In Materials and Methods (P3, L100-103), please add more details about the production process of the sparkling wine samples (i.e. the yeast used for the prise de mousse, at which temperature it was conducted…);

In Materials and Methods (P3, L110-111), authors affirm that: “Chemical compounds (9) were added to the wine samples during the disgorging process, at a concentration of 100mg/L”. How did you select the dosage of 100mg/L?   

As reported in Table 1, the only two combinations tested were: Yeast Invertase + Asparagine and Yeast Invertase + Gallic Acid. Why did not authors evaluate the combined addition of the other selected chemicals?

In Table 1, authors indicate the sample as “Manolees”, whereas in P3, L115 they talk about “Mannolees (Lallemand)”. Please check.

In order to make tables and figures readable (irrespective of the text) I suggest to clearly indicate the abbreviations (in the caption of Table 2 and in the caption of Figure 1).

In Figure 1, please indicate the units on the left Y axis. In the same Figure it is not clear what is reported on the right Y axis  

In Discussion (P8, L244), authors affirm that: “The addition of mannose was found to increase h (Fig. 2)”. Do the product Mannolees contain only mannose? Please rewrite the sentence taking into account its composition.

The conclusions of this study are very interesting. However, these results have been obtained adding chemical compounds at a dosage of 100mg/L. The application of different dosages could affect the obtained results?